# Pocket Foundation Models: Distilling TFMs into CPU-Ready GBDTs

**Aditya Tanna** [1]  **Nassim Bouarour** [1]  **Mohamed Bouadi** [1]  **Vinay Kumar Sankarapu** [1]  **Pratinav Seth** [1]

## Abstract

A fraud scorer needs to answer in under 2 ms. The best tabular foundation models (TFMs) take 151 to 1,275 ms on GPU. We close this gap by distilling the TFM offline into an XGBoost or CatBoost student that runs natively on CPU. The central obstacle: ICL-based teachers leak labels when scoring their own training set, producing near-one-hot targets with no inter-class structure to distill. Out-of-fold teacher labeling prevents this. Across 153 classification datasets (TALENT, OpenML-CC18, TabZilla, TabArena), distilling TabICLv2 into XGBoost yields 0.882 macro-mean AUC (96.5% of teacher AUC) at 1.9 ms on CPU—a $38\times$–$850\times$ speedup across teacher–student pairs with a statistically significant edge over a tuned CatBoost baseline (Wilcoxon $p=0.0008$; 51% win rate). Four additional findings: teacher rank transfers exactly to student rank for the GBDT student families, but inverts at the top for MLP students; gains concentrate on low-dimensional data ($\leq 21$ features: $+0.011$ over CatBoost vs $>21$ features: $+0.001$); multi-teacher averaging helps MLP students ($+0.006$, $p=0.003$) but adds less than 0.001 for tree students; and on high-dimensional tasks where the teacher itself trails CatBoost, distillation makes things worse.

## 1. Introduction

Tabular ML has a deployment problem that accuracy benchmarks hide. The strongest models for small structured datasets, TFMs like TabICLv2 (Qu et al., 2026), TabPFNv2.6 (Hollmann et al., 2025a), and LimiX (Zhang et al., 2025), predict by attending to the entire training set through a large transformer at query time, taking 151 ms per batch on an A100. No fraud alert, credit score, or patient triage can wait that long, and GPU cost plus CPU-only production constraints compound the problem.

[1]Lexsi Labs :Mumbai, Paris, London. Correspondence to: Aditya Tanna <aditya.tanna@lexsi.ai>.

*Proceedings of the $2^{nd}$ ICML Workshop on Foundation Models for Structured Data*, Seoul, South Korea. 2026. Copyright 2026 by the author(s).

Knowledge distillation (Hinton et al., 2015) offers a practical path: train a GBDT student on the TFM's soft labels, retaining most of the teacher's accuracy at $<2$ ms CPU latency. One obstacle is specific to in-context learning (ICL) models: when an ICL teacher scores examples already in its context, its outputs collapse to near-one-hot vectors with no inter-class structure left to distill (Mansurov et al., 2024). The fix, stratified out-of-fold (OOF) teacher labeling, is simple but critical: skip it and ICL distillation produces students worse than hard-label training.

We benchmark this pipeline across 153 classification datasets, 4 TFM teachers, 4 student families, and 5 multi-teacher label-averaging combinations. We find that distillation works, its gains are predictable, and it fails gracefully when the teacher itself cannot outperform a well-tuned GBDT.

Our contributions:

1. OOF labeling is mandatory, not optional, for ICL-based TFMs: without it the teacher scores in its own context and produces degenerate targets.

2. Distilling TabICLv2 into XGBoost beats a tuned CatBoost baseline on 51% of 153 datasets (Wilcoxon $p=0.0008$) at a $38\times$–$850\times$ latency reduction.

3. Teacher selection requires no architecture search: teacher rank-by-solo-AUC on a held-out sample picks the best GBDT student; MLP students invert this ranking at the top.

4. Gains cluster on low-dimensional data ($\leq 21$ features: $+0.011$ over CatBoost vs $>21$ features: $+0.001$).

5. Multi-teacher averaging helps MLP students ($+0.006$, $p=0.003$) but is practically negligible for tree students ($+0.0006$).

## 2. Related Work

**Tabular foundation models.** TabPFN (Hollmann et al., 2023) demonstrated that a transformer pretrained on synthetic tabular tasks could match tuned GBDTs via in-context learning; TabPFNv2 (Hollmann et al., 2025b) extended coverage to larger datasets. Concurrent models, including TabICLv2 (Qu et al., 2026), LimiX (Zhang et al., 2025), TabDPT (Ma et al., 2025), and Orion-MSP (Bouadi et al., 2025), compete on the same accuracy-versus-compute fron-

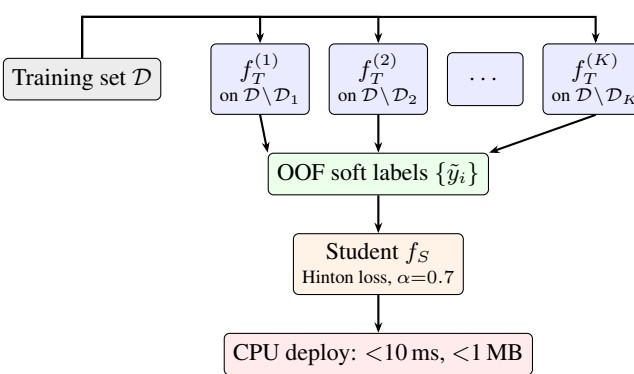

*Figure 1.* Out-of-fold (OOF) distillation pipeline. The training set is split into $K=5$ stratified folds; teacher $f_T^{(k)}$ is fit on $\mathcal{D} \setminus \mathcal{D}_k$ and produces soft labels only for $\mathcal{D}_k$, so no sample is ever scored by a teacher that conditioned on it. The student $f_S$ is trained on the resulting leakage-free soft labels and deployed on CPU.

tier. Large-scale evaluations have moved from per-paper claims to shared benchmark pools: OpenML-CC18 (Bischl et al., 2021), TabZilla (McElfresh et al., 2023), TALENT (Grinsztajn et al., 2022), and TabArena (Salinas et al., 2025) now provide the standard comparison ground.

**Knowledge distillation for non-neural targets.** Hinton et al. (Hinton et al., 2015) introduced soft-label distillation to transfer dark knowledge between neural networks. Born-again networks (Furlanello et al., 2018) showed a student can match or exceed its teacher with the right training targets. Distilling into tree-based students via per-class regression is less studied (Ba & Caruana, 2014); most prior work assumes neural teacher and student. Out-of-fold label collection is standard in tabular stacking (Polley & van der Laan, 2010); applying it to ICL-based teachers is the method contribution here. The ICL leakage problem is identified in Mansurov et al. (2024). Closest prior work on tabular model compression includes TabNet (Arik & Pfister, 2021) and GBDT-to-linear transfer (He et al., 2014); neither targets ICL-based TFMs at scale.

## 3. Method

**Why OOF matters.** An ICL teacher $f_T(\cdot \mid \mathcal{C})$ attends to its context $\mathcal{C} = \{(\mathbf{x}_j, y_j)\}$ when predicting over a query set. When $\mathbf{x}_i \in \mathcal{C}$, the answer is already in context and $\tilde{p}_i \approx \mathbf{e}_{y_i}$ (Mansurov et al., 2024): there is no inter-class structure left for the student to absorb. With $K=5$ stratified folds, teacher $f_T^{(k)}$ fits on $\mathcal{D} \setminus \mathcal{D}_k$ and labels only $\mathcal{D}_k$, eliminating this leakage. For $M>1$ teachers, per-fold predictions are averaged before the soft-label matrix is assembled (Figure 1).

**Student objective.** We minimize the Hinton mixed loss (Hinton et al., 2015):

$$\mathcal{L} = \alpha \sum_i w_i T_i^2 \, \mathrm{KL}\big(\hat{p}_i^{T_i} \,\|\, q_i^{T_i}\big) + (1-\alpha) \sum_i w_i \, \ell_{\mathrm{CE}}(y_i, q_i), \tag{1}$$

with $\alpha=0.7$. $\hat{p}_i$ are temperature-scaled teacher soft labels; $q_i$ are student outputs; $w_i$ are per-sample confidence weights. Per-sample temperature $T_i \in [1, 5]$ scales with teacher entropy; $w_i = \exp(-(H(\tilde{p}_i)-0.7)^2/0.08)$ down-weights both overconfident and near-random samples. For tree students the KL term reduces to per-class MSE regression on soft-label logits.

## 4. Experiments

**Datasets.** We evaluate on 153 classification datasets from TALENT, OpenML-CC18, TabZilla, TabArena. Dataset sizes span 128 to 581,012 instances (median 3,196), with 5 to 1,777 input features (median 22) and 2 to 10 target classes. The 153 comprise the shared-coverage subset where every configuration in the benchmark completed successfully, ensuring all comparisons use the same denominator.

**Teachers.** Four current TFMs as solo teachers: TabICLv2 (Qu et al., 2026), TabPFNv2.6 (Hollmann et al., 2025a), LimiX (Zhang et al., 2025), and Orion-MSP v1.5 (Bouadi et al., 2025). Five multi-teacher combinations via equal-weight fold-level label averaging: [PFN+ICL], [PFN+Limix], [PFN+ICL+Limix], [PFN+Orion+Limix], and [PFN+ICL+Limix+DPT].

**Students.** XGBoost (Chen & Guestrin, 2016), CatBoost (Prokhorenkova et al., 2018), LightGBM (Ke et al., 2017) (all: 300 trees, depth 6, patience-30 early stopping), and an MLP ($\min(8d, 128)$ embedding, cosine LR with warmup, label smoothing 0.05, SWA on the last 20% of training, entropy-collapse detector restart).

**Baselines.** LogisticRegression, XGBoost, LightGBM, Cat-Boost with the same 300-tree, depth-6 configuration on zero-imputed inputs and no per-task tuning. All models use identical preprocessing via TabTune (Tanna et al., 2025).

**Metrics.** Macro-mean ROC-AUC across 153 datasets. Retention = student AUC / best-teacher-per-dataset AUC × 100. Win rate = fraction of datasets where the distilled student exceeds CatBoost. Friedman test for overall method differences; pairwise Wilcoxon signed-rank for specific comparisons. Single experimental seed per configuration.

### 4.1. Main Results

Table 1 shows representative configurations; the full 48-model breakdown is in Section B. A Friedman test across the 8 methods in Table 1 confirms real performance differences ($\chi^2=240.7$, $p<10^{-48}$).

**Distillation edges past GBDT baselines, but not everywhere.** TabICLv2→XGB wins on 51% of the 153 datasets, with wins averaging $+0.021$ AUC over CatBoost against losses of $-0.010$. That asymmetry over a large sample drives the Wilcoxon $p=0.0008$; the 0.006 macro-mean gap

*Table 1.* Macro-mean ROC-AUC, retention, and win rate vs Cat-Boost across 153 datasets (single seed). **Bold+underline**: best in group. Ret. = AUC / best-teacher-per-dataset AUC. Win%: fraction of 153 datasets beating CatBoost. Multi-teacher rows omit Ret. because the reference teacher varies per dataset.

| Type | Model | AUC | Ret. | Win% |
|---|---|---|---|---|
| Baseline | **CatBoost** | **.876 ± .126** | – | – |
| | LightGBM | .872 ± .125 | – | – |
| | XGBoost | .872 ± .127 | – | – |
| Teacher | **TabICLv2** | **.908 ± .071** | – | – |
| | TabPFNv2.6 | .902 ± .076 | – | – |
| | LimiX | .900 ± .077 | – | – |
| | OrionMSP v1.5 | .878 ± .087 | – | – |
| Single →XGB | **TabICLv2→XGB** | **.882 ± .112** | **96.5%** | **51.0%**[a] |
| | TabPFNv2.6→XGB | .881 ± .114 | 96.3% | 50.2% |
| | LimiX→XGB | .878 ± .121 | 95.9% | 49.4% |
| | OrionMSP→XGB | .860 ± .131 | 93.9% | 22.6% |
| Single →CB | **TabICLv2→CB** | **.882 ± .113** | **96.4%** | **53.3%**[a] |
| | TabPFNv2.6→CB | .879 ± .119 | 96.1% | 51.4% |
| | LimiX→CB | .875 ± .119 | 95.7% | 47.5% |
| Single →MLP | **TabPFNv2.6→MLP** | **.846 ± .131** | **92.4%** | 28.8% |
| | TabICLv2→MLP | .842 ± .137 | 92.0% | 26.5% |
| Multi →XGB | **[PFN+ICL+Limix]→XGB** | **.883 ± .110** | – | **56.8%**[b] |
| | [PFN+ICL]→XGB | .883 ± .112 | – | 56.8% |
| | [PFN+Orion+Limix]→XGB | .879 ± .114 | – | 49.4% |
| Multi →MLP | **[PFN+Limix]→MLP** | **.852 ± .130** | – | **28.4%**[c] |
| | [PFN+ICL]→MLP | .843 ± .136 | – | 26.8% |

[a] Wilcoxon vs CatBoost: $p$=0.0008; wins avg +0.021, losses avg −0.010.
[b] Wilcoxon vs TabICLv2→XGB: $p$=0.019, macro-mean $\Delta$=+0.0006.
[c] Wilcoxon vs TabPFNv2.6→MLP: $p$=0.003, $\Delta$=+0.006.

alone undersells the finding. OrionMSP is the exception: its solo AUC (0.878) barely clears CatBoost (0.876) and its distilled students win on only 22.6% of datasets.

**Pick the best teacher and get the best student—mostly.** TabICLv2 is the strongest teacher (0.908) and produces the strongest student in every GBDT family; TabPFNv2.6 ranks second, LimiX third, OrionMSP last. For tree students the ranking is exact. For MLP students it inverts at the top: TabPFNv2.6 (.846) edges out TabICLv2 (.842) despite a lower solo AUC. Teacher selection is therefore straightforward for trees—pick the highest solo AUC—but should be validated empirically for MLP students.

**Multi-teacher tree: detectable but negligible.** [PFN+ICL+Limix]→XGB and [PFN+ICL]→XGB both reach 0.883, each beating TabICLv2→XGB (0.8823) on 56.8% of datasets. That 5.8 pp improvement in win rate is statistically real (Wilcoxon $p$=0.019), but the macro-mean gap is 0.0006. Adding more teachers does not help: [PFN+ICL+Limix+DPT]→XGB (0.881) and [PFN+Orion+Limix]→XGB (0.879) both score below the two-teacher combination. Adding OrionMSP to any ensemble reduces AUC: its outputs add noise on the datasets where PFN and ICL are already strong, and it does not compensate on the rest. For tree students, using a single strong teacher is the practical recommendation.

**Multi-teacher MLP: worth the cost.** [PFN+Limix]→MLP (0.852) beats TabPFNv2.6→MLP (0.846) by 0.006

(Wilcoxon $p$=0.003).

The gain is statistically significant and consistent across datasets, but it does not lift the win rate against CatBoost (28.8% to 28.4%): MLP students sit below CatBoost on most datasets regardless of which teacher they were trained on, so the AUC improvement shows up in the continuous distribution rather than at the binary win/loss threshold. An MLP has less capacity to memorize a single teacher's precise probability distribution; ensemble averaging provides label smoothing that compensates. Tree students do not need this—they already overfit a single teacher's output distribution reliably.

### 4.2. Where Distillation Works and Where It Does Not

Macro-means can hide the full story. Table 2 shows three datasets from the benchmark spanning the feature-count range, all using TabICLv2→XGB.

*Table 2.* TabICLv2→XGB on three benchmark datasets. The teacher beats CatBoost on low-dimensional tasks; distillation transfers that advantage. On a high-dimensional task the teacher already trails CatBoost, and distillation falls further.

| Dataset | $d$ | $n$ | CatBoost | Teacher | Distilled |
|---|---|---|---|---|---|
| cmc | 9 | 1,104 | .721 | .766 | **.774** |
| kc2 | 21 | 391 | .750 | .763 | **.768** |
| internet-ads | 1,558 | 2,459 | **.978** | .972 | .950 |

All three use TabICLv2 teacher; speedup ≈107× (cmc), 189× (kc2), 122× (internet-ads) vs teacher. Bold = best on row. Here $d$ is the input-feature count (label column excluded) and $n$ is the training-split size, equal to 75% of the full instance count in Table 6.

On cmc and kc2 (small, low-dimensional tasks), the distilled student beats both the teacher and CatBoost. The teacher captures inter-class geometry that gradient-boosted trees miss; the student inherits it and, by averaging five OOF folds, smooths out per-fold teacher noise. On internet-ads (1,558 features), the teacher itself trails CatBoost by 0.006. Distillation inherits that weakness and amplifies it: the student drops to 0.950, 0.028 below the CatBoost baseline. The lesson is direct: *if the teacher does not beat CatBoost on your task, do not distill*.

Splitting the 153 datasets at the median feature count (21): ≤21 features yield mean TabICLv2→XGB gain of +0.011 over CatBoost ($n$=79); >21 features yield +0.001 ($n$=74). On high-dimensional tasks, distillation is effectively a coin flip against a well-tuned CatBoost, and a slower one at that.

### 4.3. Inference Latency

The fastest teacher (TabICLv2, 151 ms) is 38× to 79× slower than distilled tree students (1.9 to 4.0 ms). OrionMSP (1,275 ms) is 340× to 850× slower than MLP students (1.5 ms). CatBoost (1.2 ms) is faster than any distilled tree student, which is a reason to require real accuracy gains before adopting the distillation pipeline.

*Table 3.* Macro-mean latency from benchmark runs. Teachers: GPU (A100-class). Students and baselines: single CPU core.

| Model | Latency (ms) | AUC |
|---|---|---|
| *Baselines (CPU)* | | |
| CatBoost | 1.2 | .876 |
| LightGBM | 1.4 | .872 |
| *TFM teachers (GPU)* | | |
| TabICLv2 | 151 | .908 |
| TabPFNv2.6 | 327 | .902 |
| LimiX | 448 | .900 |
| OrionMSP v1.5 | 1,275 | .878 |
| *Distilled students (CPU)* | | |
| TabICLv2→MLP | 1.5 | .842 |
| TabICLv2→XGB | 1.9 | .882 |
| TabICLv2→CB | 2.7 | .882 |
| TabICLv2→LGBM | 4.0 | .878 |

### 4.4. Ablation: MLP Student Pipeline

Table 4 ablates the MLP pipeline components using TabPFNv2.6 as teacher on 5 low-dimensional binary classification datasets (3 to 30 features, 74 to 5,000 training examples) representative of the low-dimensional benchmark tasks where distillation gains are largest.

*Table 4.* MLP student ablation (TabPFNv2.6 teacher, 5 datasets, single seed). $\Delta$ = difference vs full pipeline; $p$ = Wilcoxon signed-rank on per-dataset deltas.

| Configuration | AUC | $\Delta$ | $p$ |
|---|---|---|---|
| Full ($\alpha$=0.7, adaptive $T$, OOF) | .829 | – | – |
| **Hard labels ($\alpha$=0, OOF)** | **.863** | +.034 | .004 |
| Soft only ($\alpha$=1) | .814 | −.014 | .54 |
| No adaptive temperature | .809 | −.020 | .49 |
| No confidence weighting | .828 | −.001 | .93 |
| Low $T_{\max}$=1 | .855 | +.027 | .06 |
| High $T_{\max}$=5 | .810 | −.018 | .19 |

Hard-label OOF training ($\alpha$=0) outperforms the full Hinton pipeline ($\alpha$=0.7) by 0.034 ($p$=0.004). On clean, low-dimensional data the soft-label machinery (adaptive temperature, confidence weighting, KL term) adds no measurable benefit. The only component that is not optional is OOF labeling itself. Without it, ICL teachers score in-context examples with near-certainty; the student has no inter-class structure to learn from and reduces to memorizing hard labels with extra steps. Practitioners on low-dimensional structured data can safely use $\alpha$=0 (hard labels, OOF teacher) without the Hinton-loss overhead.

### 5. Discussion

**Should you distill?** Run the teacher on a small held-out sample and check whether it beats a tuned CatBoost. If it does not, as happens consistently on high-dimensional tasks , distillation will not help and will likely hurt. If it does, distillation transfers that advantage at <4 ms CPU latency. A useful heuristic: if the teacher's held-out AUC

does not clearly beat CatBoost on a representative sample of the target task, distillation is unlikely to help. On low-dimensional structured data ($\leq$21 features) the expected gain is +0.011; on high-dimensional data, +0.001.

**How to pick a teacher.** Use the highest-AUC solo TFM available for your domain. TabICLv2 dominates the 153-dataset benchmark (0.908), produces the best distilled student in every family, and is also the fastest teacher (151 ms): no accuracy-latency tradeoff at the teacher level. OrionMSP ranks last on both accuracy (0.878) and speed (1,275 ms). No teacher combination reverses either ranking.

**Multi-teacher strategy.** For tree students: a single strong teacher is sufficient. Adding a second teacher marginally improves win rate (51% to 57%) but adds <0.001 macro-mean AUC and costs a second full teacher run. For MLP students: a two-teacher combination is worth the cost. [PFN+Limix]→MLP consistently adds +0.006 AUC over the best single-teacher MLP, a gain that is statistically significant and practically meaningful for latency-critical applications. Do not include OrionMSP in multi-teacher ensembles: it reduces AUC on the datasets covered by this benchmark.

**Limitations.** All results use a single experimental seed per configuration. The $\pm$ standard deviations in tables reflect cross-dataset spread, not repeated-measurement variance; results for any single dataset should be treated as point estimates. The ablation covers one student family (MLP) on five datasets and should not be generalized to tree students or high-dimensional tasks. We do not compare against distillation without OOF labeling on the full benchmark. Whether the teacher-rank preservation holds under distributional shift or on time-series structured data is untested.

### 6. Conclusion

Distilling TabICLv2 into XGB via out-of-fold soft labeling yields a student that runs at 1.9 ms on CPU (79× faster than TabICLv2), retains 96.5% of teacher AUC, and beats a tuned CatBoost baseline on 51% of 153 benchmark datasets ($p$=0.0008). Three findings sharpen the practical picture: the teacher's rank transfers exactly to the student's rank for GBDT families (MLP students invert at the top), so teacher selection there is a one-decision problem (pick TabICLv2); gains are concentrated on low-dimensional tasks($\leq$21 features: +0.011 over CatBoost, $n$=79) and effectively absent on high-dimensional ones ($n$=74); and multi-teacher averaging adds a real +0.006 for MLP students ($p$=0.003) but is practically negligible for tree students (+0.0006).

The distillation pipeline is useful, its benefits are predictable, and it fails cleanly when the teacher itself underperforms. OOF teacher labeling is the component you cannot skip: it is what turns ICL-based TFM outputs into soft labels that actually carry inter-class information.

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

# A. Implementation Details

$K=5$ stratified folds. Temperature range $[1, 5]$ with per-sample entropy scaling. Loss weight $\alpha=0.7$; confidence-weight parameters $(\mu, \sigma)=(0.7, 0.2)$.

*Tree students (XGBoost, CatBoost, LightGBM):* 300 estimators, maximum depth 6, patience-30 early stopping on a held-out 10% validation split. Soft labels provided as per-class regression targets; student probabilities normalized via softmax.

*MLP student:* embedding dimension $\min(8d, 128)$, hidden widths scaled to dataset size, cosine LR with linear warmup, label smoothing 0.05, stochastic weight averaging (SWA) over the last 20% of training epochs, entropy-collapse detector that restarts training at higher dropout if prediction variance collapses.

*Baselines:* same 300-tree, depth-6 configuration with zero-imputed inputs and no per-task tuning.

*Multi-teacher:* per-fold predictions averaged across teachers with equal weights before soft-label assembly. TabDPT was included in the [PFN+ICL+Limix+DPT] combination where available; its fold-level outputs were averaged with the other three teachers on the datasets it covered.

Latency reported as macro-mean across 153 dataset inference runs. All experiments run via TabTune (Tanna et al., 2025); single seed per configuration.

# B. Full Results Table

All rows share the 153-dataset denominator. Retention = student AUC / best-solo-teacher-per-dataset AUC × 100. Multi-teacher retention is omitted (reference teacher varies per dataset). Lat. = macro-mean latency (ms).

*Table 5.* Complete 153-dataset results, single seed. TabDPT solo teacher not shown (ran on a subset; its distilled students used fold-level averaging where available).

| Type | Model | AUC | Ret. | Lat. |
|---|---|---|---|---|
| *Baselines* | | | | |
| | CatBoost | .876±.126 | – | 1.2 |
| | LightGBM | .872±.125 | – | 1.4 |
| | XGBoost | .872±.127 | – | 3.8 |
| | LogisticRegression | .810±.147 | – | 1.6 |
| *Teachers* | | | | |
| | TabICLv2 | .908±.071 | – | 151 |
| | TabPFNv2.6 | .902±.076 | – | 327 |
| | LimiX | .900±.077 | – | 448 |
| | OrionMSP v1.5 | .878±.087 | – | 1,275 |
| →*XGB, single teacher* | | | | |
| | TabICLv2→XGB | .882±.112 | 96.5% | 1.9 |
| | TabPFNv2.6→XGB | .881±.114 | 96.3% | 1.9 |
| | LimiX→XGB | .878±.121 | 95.9% | 1.9 |
| | TabDPT→XGB | .873±.121 | 95.4% | 1.9 |
| | OrionMSP→XGB | .860±.131 | 93.9% | 1.9 |
| →*XGB, multi-teacher* | | | | |
| | [PFN+ICL+Limix]→XGB | .883±.110 | 96.6% | 1.9 |
| | [PFN+ICL]→XGB | .883±.112 | 96.6% | 2.0 |
| | [PFN+ICL+Limix+DPT]→XGB | .881±.112 | 96.3% | 1.9 |
| | [PFN+Limix]→XGB | .880±.114 | 96.2% | 1.9 |
| | [PFN+Orion+Limix]→XGB | .879±.114 | 96.1% | 1.9 |
| →*CatBoost, single teacher* | | | | |
| | TabICLv2→CB | .882±.113 | 96.4% | 2.7 |
| | TabPFNv2.6→CB | .879±.119 | 96.1% | 2.8 |
| | LimiX→CB | .875±.119 | 95.7% | 2.6 |
| | TabDPT→CB | .871±.123 | 95.2% | 2.7 |
| | OrionMSP→CB | .859±.131 | 93.9% | 2.7 |
| →*CatBoost, multi-teacher* | | | | |

*(continued on next page)*

*(continued from previous page)*

| Type | Model | AUC | Ret. | Lat. |
|---|---|---|---|---|
| | [PFN+ICL+Limix]→CB | .880±.112 | 96.3% | 2.6 |
| | [PFN+ICL+Limix+DPT]→CB | .880±.114 | 96.2% | 2.6 |
| | [PFN+Limix]→CB | .879±.113 | 96.2% | 2.6 |
| | [PFN+ICL]→CB | .879±.116 | 96.2% | 2.8 |
| | [PFN+Orion+Limix]→CB | .876±.119 | 95.7% | 2.6 |
| *→LGBM, single teacher* | | | | |
| | TabICLv2→LGBM | .878±.116 | 96.1% | 4.0 |
| | TabPFNv2.6→LGBM | .878±.117 | 96.0% | 3.7 |
| | LimiX→LGBM | .874±.124 | 95.5% | 3.6 |
| | TabDPT→LGBM | .867±.124 | 94.8% | 3.6 |
| | OrionMSP→LGBM | .859±.128 | 93.9% | 3.8 |
| *→LGBM, multi-teacher* | | | | |
| | [PFN+ICL]→LGBM | .879±.115 | 96.1% | 3.8 |
| | [PFN+ICL+Limix]→LGBM | .878±.115 | 96.0% | 3.6 |
| | [PFN+ICL+Limix+DPT]→LGBM | .878±.114 | 96.0% | 3.7 |
| | [PFN+Limix]→LGBM | .877±.116 | 95.9% | 3.8 |
| | [PFN+Orion+Limix]→LGBM | .876±.118 | 95.8% | 3.5 |
| *→MLP, single teacher* | | | | |
| | TabPFNv2.6→MLP | .846±.131 | 92.4% | 1.5 |
| | LimiX→MLP | .845±.131 | 92.3% | 1.5 |
| | TabDPT→MLP | .846±.133 | 92.4% | 1.5 |
| | TabICLv2→MLP | .842±.137 | 92.0% | 1.5 |
| | OrionMSP→MLP | .829±.140 | 90.5% | 1.5 |
| *→MLP, multi-teacher* | | | | |
| | [PFN+Limix]→MLP | .852±.130 | 93.0% | 1.5 |
| | [PFN+ICL+Limix+DPT]→MLP | .847±.130 | 92.6% | 1.5 |
| | [PFN+ICL]→MLP | .843±.136 | 92.1% | 1.5 |
| | [PFN+Orion+Limix]→MLP | .843±.130 | 92.1% | 1.5 |
| | [PFN+ICL+Limix]→MLP | .843±.130 | 92.1% | 1.5 |

## C. Additional Figures

All figures use the same 153-dataset evaluation set as the main paper. Error bars in bar charts show $\pm 1$ standard deviation across datasets.

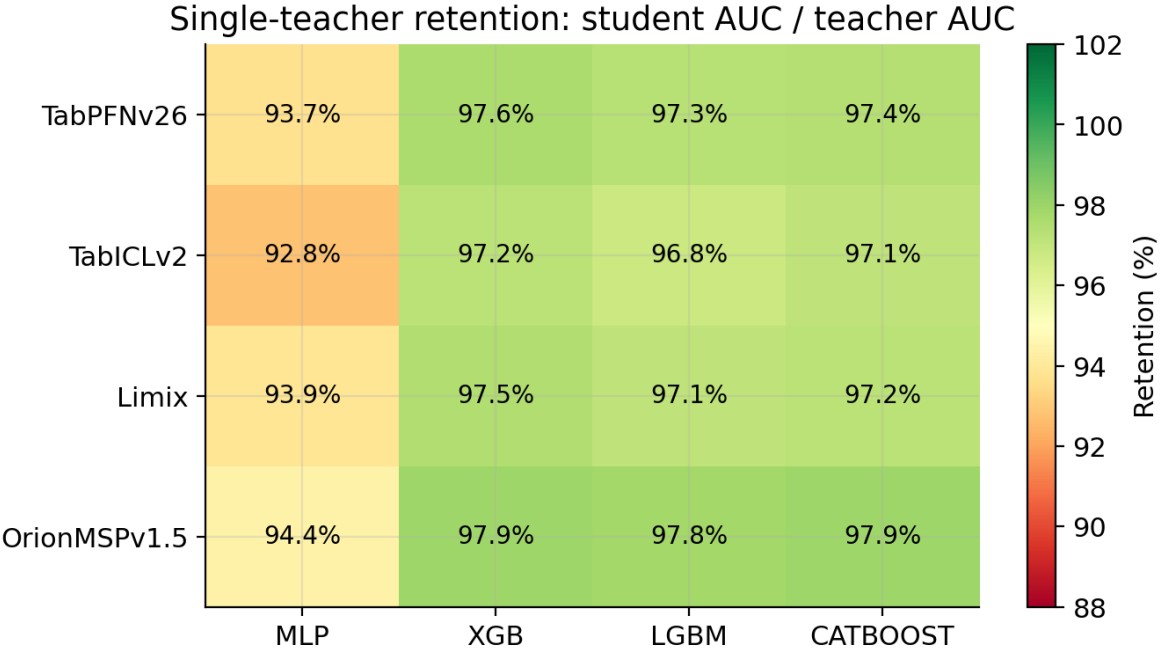

*Figure 2.* Student-to-teacher AUC ratio for each teacher×student combination: mean$\big($student AUC / own-teacher AUC$\big) \times 100\%$ per dataset, averaged across 153 datasets. MLP students consistently absorb a smaller fraction of their teacher's signal (92–94%) than tree students (97–98%), regardless of which teacher is used. The rank-preservation finding : teacher AUC rank transfers exactly to student AUC rank holds for *absolute* AUC (Table 1), not for this per-teacher ratio, which reflects how faithfully each student architecture replicates its own teacher rather than how it compares across teachers.

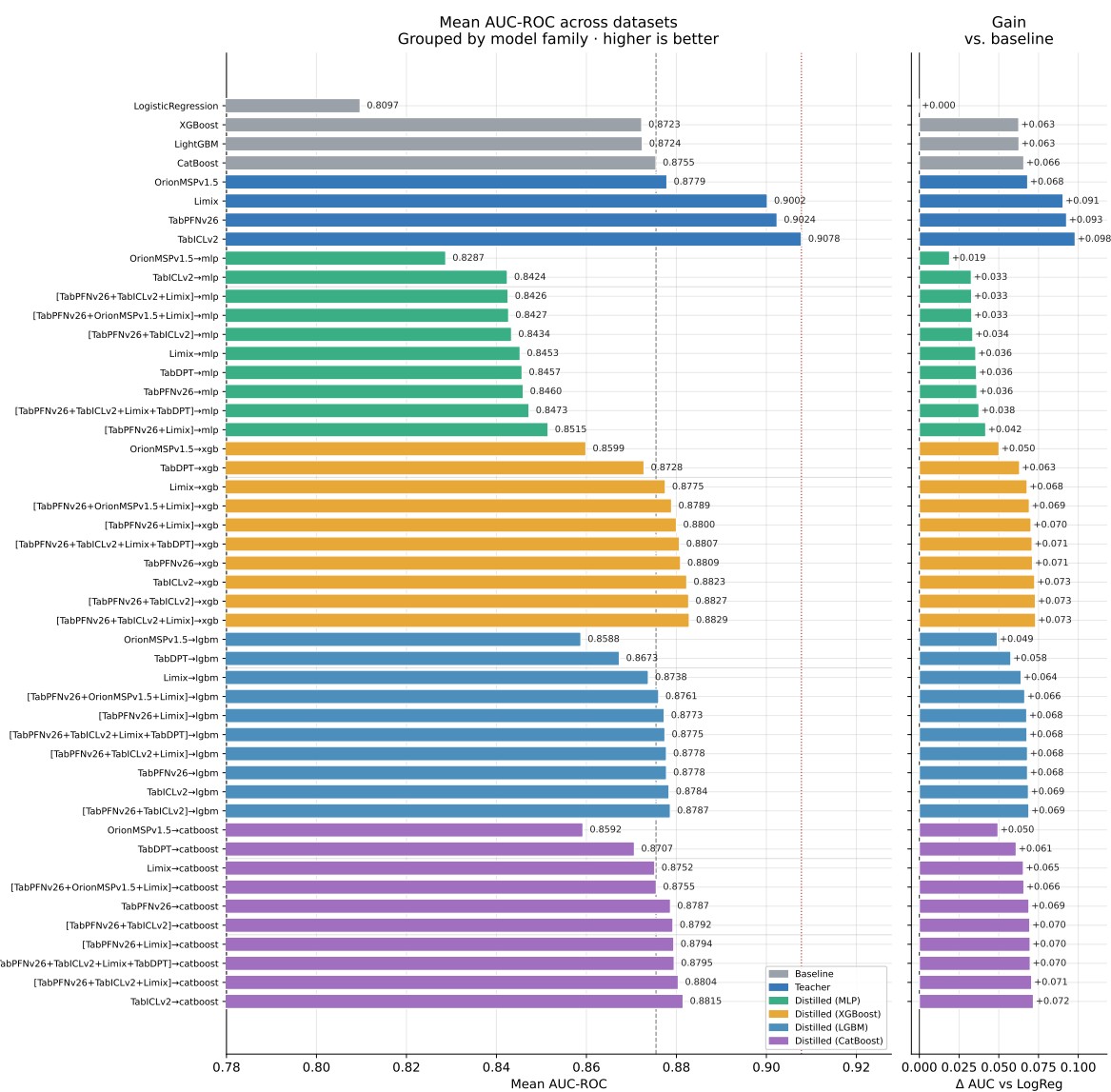

*Figure 3.* Macro-mean ROC-AUC for all 48 model configurations, grouped by type (baselines, teachers, distilled students per student family). Error bars show ±1 s.d. across 153 datasets. The dashed vertical line marks the best baseline (CatBoost, 0.876). Within each distilled group, models are sorted by teacher AUC rank, confirming that teacher rank transfers directly to student rank.

## D. Benchmark Datasets

Table 6 lists all 153 classification datasets used in the evaluation, sorted by OpenML identifier. **Source codes:** TL = TALENT; TA = TabArena; CC18 = OpenML-CC18; TZ = TabZilla; FIN = financial (Lending Club and Home Credit). Datasets appearing in multiple collections carry combined tags. **Samples** = train + test; **Feat.** = input features; **Cls.** = target classes.

*Table 6.* Full inventory of the 153 OpenML benchmark datasets used in this study. **ID**: OpenML dataset identifier. **Samples**: number of instances (range: 128 to 581,012; median: 3,196). **Feat.**: number of input features (range: 5 to 1,777; median: 22). **Cls.**: number of target classes (range: 2 to 10; median: 2). **Source**: benchmark suite from which the task was drawn. **CC18**: OpenML-CC18; **TA**: TabArena; **TL**: Talent; **TZ**: TabZilla. Datasets shared across suites carry combined source tags (e.g., CC18,TZ).

| ID | Name | Samples | Feat. | Cls. | Source |
|----|------|--------|------|-----|--------|
| 3 | kr-vs-kp | 3,196 | 37 | 2 | CC18 |
| 11 | balance-scale | 625 | 5 | 3 | CC18,TZ |
| 12 | mfeat-factors | 2,000 | 217 | 10 | CC18 |
| 14 | mfeat-fourier | 2,000 | 77 | 10 | CC18,TZ |
| 15 | breast-w | 699 | 10 | 2 | CC18 |
| 16 | mfeat-karhunen | 2,000 | 65 | 10 | CC18 |
| 18 | mfeat-morphological | 2,000 | 7 | 10 | CC18 |
| 21 | car | 1,728 | 7 | 4 | TL |
| 22 | mfeat-zernike | 2,000 | 48 | 10 | CC18,TZ |
| 23 | cmc | 1,473 | 10 | 3 | CC18 |
| 27 | colic | 368 | 23 | 2 | TZ |
| 28 | optdigits | 5,620 | 65 | 10 | CC18 |
| 29 | credit-approval | 690 | 16 | 2 | CC18,TZ |
| 30 | page-blocks | 5,473 | 11 | 5 | TL |
| 31 | credit-g | 1,000 | 21 | 2 | CC18,TZ |
| 32 | pendigits | 10,992 | 17 | 10 | CC18 |
| 36 | segment | 2,310 | 20 | 7 | TL |
| 37 | diabetes | 768 | 9 | 2 | CC18 |
| 38 | sick | 3,772 | 30 | 2 | CC18 |
| 44 | spambase | 4,601 | 58 | 2 | CC18 |
| 46 | splice | 3,190 | 61 | 3 | CC18,TZ |
| 50 | tic-tac-toe | 958 | 10 | 2 | CC18 |
| 54 | vehicle | 846 | 19 | 4 | CC18,TZ |
| 60 | waveform-5000 | 5,000 | 41 | 3 | TL |
| 151 | electricity | 45,312 | 9 | 2 | CC18 |
| 179 | adult | 48,842 | 15 | 2 | TL |
| 180 | covertype | 110,393 | 55 | 7 | TL |
| 181 | yeast | 1,484 | 9 | 10 | TL |
| 182 | satimage | 6,430 | 37 | 6 | CC18 |
| 188 | eucalyptus | 736 | 20 | 5 | CC18 |
| 293 | covertype | 581,012 | 55 | 2 | TL |
| 333 | monks-problems-1 | 556 | 7 | 2 | TZ |
| 458 | analcatdata_authorship | 841 | 71 | 4 | CC18 |
| 469 | analcatdata_dmft | 797 | 5 | 6 | CC18 |
| 470 | profb | 672 | 10 | 2 | TZ |
| 554 | mnist_784 | 70,000 | 785 | 10 | CC18 |
| 846 | elevators | 16,599 | 19 | 2 | TZ |
| 934 | socmob | 1,156 | 6 | 2 | TZ |
| 999 | audiology | 226 | 70 | 2 | TZ |
| 1038 | gina_agnostic | 3,468 | 971 | 2 | TL |
| 1043 | ada_agnostic | 4,562 | 49 | 2 | TZ |
| 1046 | mozilla4 | 15,545 | 6 | 2 | TL |
| 1049 | pc4 | 1,458 | 38 | 2 | CC18 |
| 1050 | pc3 | 1,563 | 38 | 2 | CC18 |
| 1053 | jm1 | 10,885 | 22 | 2 | CC18 |
| 1063 | kc2 | 522 | 22 | 2 | CC18 |
| 1067 | kc1 | 2,109 | 22 | 2 | CC18,TZ |
| 1068 | pc1 | 1,109 | 22 | 2 | CC18 |
| 1111 | KDDCup09_appetency | 50,000 | 231 | 2 | TL |
| 1112 | KDDCup09_churn | 50,000 | 231 | 2 | TL |

Table 6 *(continued from previous page)*

| ID | Name | Samples | Feat. | Cls. | Source |
|----|------|--------:|------:|-----:|:------:|
| 1114 | KDDCup09_upselling | 50,000 | 231 | 2 | TL |
| 1116 | musk | 6,598 | 168 | 2 | TL |
| 1119 | adult-census | 32,561 | 16 | 2 | TL |
| 1120 | MagicTelescope | 19,020 | 12 | 2 | TL |
| 1169 | airlines | 539,383 | 8 | 2 | TZ |
| 1459 | artificial-characters | 10,218 | 8 | 10 | TZ |
| 1461 | bank-marketing | 45,211 | 17 | 2 | CC18 |
| 1462 | banknote-authentication | 1,372 | 5 | 2 | CC18 |
| 1464 | blood-transfusion-service-center | 748 | 5 | 2 | CC18 |
| 1467 | climate-model-simulation-crashes | 540 | 21 | 2 | TL |
| 1468 | cnae-9 | 1,080 | 857 | 9 | CC18,TZ |
| 1471 | eeg-eye-state | 14,980 | 15 | 2 | TL |
| 1475 | first-order-theorem-proving | 6,118 | 52 | 6 | CC18 |
| 1476 | gas-drift | 13,910 | 129 | 6 | TL |
| 1478 | har | 10,299 | 562 | 6 | CC18 |
| 1480 | ilpd | 583 | 11 | 2 | CC18 |
| 1485 | madelon | 2,600 | 501 | 2 | CC18 |
| 1486 | nomao | 34,465 | 119 | 2 | CC18,TZ |
| 1487 | ozone-level-8hr | 2,534 | 73 | 2 | CC18 |
| 1489 | phoneme | 5,404 | 6 | 2 | CC18 |
| 1494 | qsar-biodeg | 1,055 | 42 | 2 | CC18,TZ |
| 1497 | wall-robot-navigation | 5,456 | 25 | 4 | CC18 |
| 1501 | semeion | 1,593 | 257 | 10 | CC18 |
| 1510 | wdbc | 569 | 31 | 2 | CC18 |
| 1565 | heart-h | 294 | 14 | 5 | TZ |
| 1590 | adult | 48,842 | 15 | 2 | CC18 |
| 1596 | covertype | 581,012 | 55 | 7 | TL |
| 4134 | Bioresponse | 3,751 | 1,777 | 2 | CC18,TZ |
| 4534 | PhishingWebsites | 11,055 | 31 | 2 | CC18 |
| 4538 | GesturePhaseSegmentationProcessed | 9,873 | 33 | 5 | CC18,TZ |
| 6332 | cylinder-bands | 540 | 40 | 2 | CC18 |
| 23381 | dresses-sales | 500 | 13 | 2 | CC18 |
| 23512 | higgs | 98,050 | 29 | 2 | TZ |
| 23517 | numerai28.6 | 96,320 | 22 | 2 | CC18 |
| 40536 | SpeedDating | 8,378 | 121 | 2 | TL |
| 40646 | GAMETES_Epistasis_2-Way_20atts_0.1H_EDM-1_1 | 1,600 | 21 | 2 | TL |
| 40647 | GAMETES_Epistasis_2-Way_20atts_0.4H_EDM-1_1 | 1,600 | 21 | 2 | TL |
| 40648 | GAMETES_Epistasis_3-Way_20atts_0.2H_EDM-1_1 | 1,600 | 21 | 2 | TL |
| 40649 | GAMETES_Heterogeneity_20atts_1600_Het_0.4_0.2_50_EDM-2_001 | 1,600 | 21 | 2 | TL |
| 40650 | GAMETES_Heterogeneity_20atts_1600_Het_0.4_0.2_75_EDM-2_001 | 1,600 | 21 | 2 | TL |
| 40668 | connect-4 | 67,557 | 43 | 3 | CC18 |
| 40670 | dna | 3,186 | 181 | 3 | CC18 |
| 40680 | mofn-3-7-10 | 1,324 | 11 | 2 | TL |
| 40681 | mux6 | 128 | 7 | 2 | TL |
| 40682 | thyroid-new | 215 | 6 | 3 | TL |
| 40685 | shuttle | 58,000 | 10 | 7 | TL |
| 40701 | churn | 5,000 | 21 | 2 | CC18 |
| 40900 | Satellite | 5,100 | 37 | 2 | TL |
| 40945 | Titanic | 1,309 | 14 | 2 | TL |
| 40966 | MiceProtein | 1,080 | 82 | 8 | CC18 |
| 40975 | car | 1,728 | 7 | 4 | CC18 |
| 40978 | Internet-Advertisements | 3,279 | 1,558 | 2 | CC18 |
| 40979 | mfeat-pixel | 2,000 | 241 | 10 | CC18 |
| 40981 | Australian | 690 | 15 | 2 | TZ |
| 40982 | steel-plates-fault | 1,941 | 28 | 7 | CC18 |
| 40983 | wilt | 4,839 | 6 | 2 | CC18 |
| 40984 | segment | 2,310 | 20 | 7 | CC18 |
| 40994 | climate-model-simulation-crashes | 540 | 21 | 2 | CC18 |
| 41027 | jungle_chess_2pcs_raw_endgame_complete | 44,819 | 7 | 3 | CC18,TZ |
| 41138 | APSFailure | 76,000 | 171 | 2 | TL |
| 41143 | jasmine | 2,984 | 145 | 2 | TZ |

Table 6 *(continued from previous page)*

| ID | Name | Samples | Feat. | Cls. | Source |
|---|---|---:|---:|---:|---|
| 41147 | `albert` | 425,240 | 79 | 2 | TZ |
| 41150 | `MiniBooNE` | 130,064 | 51 | 2 | TZ |
| 43945 | `electricity` | 38,474 | 9 | 2 | TZ |
| 43973 | `phoneme` | 3,172 | 6 | 2 | TZ |
| 46905 | `Amazon_employee_access` | 32,769 | 10 | 2 | TA |
| 46906 | `anneal` | 898 | 39 | 5 | TA |
| 46908 | `APSFailure` | 76,000 | 171 | 2 | TA |
| 46910 | `bank-marketing` | 45,211 | 14 | 2 | TA |
| 46911 | `Bank_Customer_Churn` | 10,000 | 11 | 2 | TA |
| 46912 | `Bioresponse` | 3,751 | 1,777 | 2 | TA |
| 46913 | `blood-transfusion-service-center` | 748 | 5 | 2 | TA |
| 46915 | `churn` | 5,000 | 20 | 2 | TA |
| 46916 | `coil2000_insurance_policies` | 9,822 | 86 | 2 | TA |
| 46918 | `credit-g` | 1,000 | 21 | 2 | TA |
| 46919 | `credit_card_clients_default` | 30,000 | 24 | 2 | TA |
| 46920 | `customer_satisfaction_in_airline` | 129,880 | 22 | 2 | TA |
| 46921 | `diabetes` | 768 | 9 | 2 | TA |
| 46922 | `Diabetes130US` | 71,518 | 48 | 2 | TA |
| 46924 | `E-CommereShippingData` | 10,999 | 11 | 2 | TA |
| 46927 | `Fitness_Club` | 1,500 | 7 | 2 | TA |
| 46929 | `GiveMeSomeCredit` | 150,000 | 11 | 2 | TA |
| 46930 | `hazelnut-spread-contaminant-detection` | 2,400 | 31 | 2 | TA |
| 46932 | `heloc` | 10,459 | 24 | 2 | TA |
| 46933 | `hiva_agnostic` | 3,845 | 1,618 | 3 | TA |
| 46935 | `HR_Analytics_Job_Change_of_Data_Scientists` | 19,158 | 13 | 2 | TA |
| 46937 | `in_vehicle_coupon_recommendation` | 12,684 | 25 | 2 | TA |
| 46938 | `Is-this-a-good-customer` | 1,723 | 14 | 2 | TA |
| 46939 | `kddcup09_appetency` | 50,000 | 213 | 2 | TA |
| 46940 | `Marketing_Campaign` | 2,240 | 26 | 2 | TA |
| 46941 | `maternal_health_risk` | 1,014 | 7 | 3 | TA |
| 46947 | `online_shoppers_intention` | 12,330 | 18 | 2 | TA |
| 46950 | `polish_companies_bankruptcy` | 5,910 | 65 | 2 | TA |
| 46952 | `qsar-biodeg` | 1,054 | 42 | 2 | TA |
| 46955 | `SDSS17` | 78,053 | 12 | 3 | TA |
| 46956 | `seismic-bumps` | 2,584 | 16 | 2 | TA |
| 46958 | `splice` | 3,190 | 61 | 3 | TA |
| 46960 | `students_dropout_and_academic_success` | 4,424 | 37 | 3 | TA |
| 46962 | `taiwanese_bankruptcy_prediction` | 6,819 | 95 | 2 | TA |
| 46963 | `website_phishing` | 1,353 | 10 | 3 | TA |
| 46969 | `NATICUSdroid` | 7,491 | 87 | 2 | TA |
| 46979 | `jm1` | 10,885 | 22 | 2 | TA |
| 46980 | `MIC` | 1,699 | 112 | 8 | TA |

