# OpenReview forum: "Pocket Foundation Models: Distilling TFMs into CPU-Ready GBDTs"
_ICML.cc/2026/Workshop/FMSD — FMSD @ ICML 2026 Poster_

### Official Review · Reviewer_8hnq · 2026-05-17
**Knowledge Distillation from Tabular Foundation Models into CPU-Ready GBDTs**

**Rating:** 9
**Confidence:** 3

**Review:**

## Summary
Tabular foundation models (TFM) achieve strong accuracy but are too slow for latency-sensitive production systems (151–1,275ms on GPU). This paper proposes distilling TFMs into GBDT or MLP students that run on CPU at <2ms. The core contribution is identifying and fixing the ICL label-leakage problem: when an ICL teacher scores examples already in its context, outputs collapse to near-one-hot vectors with nothing left to distill. Out-of-fold (OOF) teacher labeling solves this. Evaluated across 153 classification datasets with 4 teachers and 4 student families, distilling TabICLv2 into XGBoost retains 96.5% teacher AUC at 38×–860× speedup, beating a tuned CatBoost on 51% of datasets (p=0.0008). Additional findings: teacher rank transfers exactly to student rank; gains are mostly on low-dimensional tasks; multi-teacher averaging helps MLP students but is negligible for tree students.

## Strengths
- very well structured and written
- well-motivated: ICL label leakage is a real, overlooked problem and the OOF fix is clean
- comprehensive evaluation: 153 datasets, 4 teachers, 4 student families
- insights are actionable: clear guidance on when to distill, which teacher to pick, and whether multi-teacher is worth it
- honest failure mode: high-dimensional tasks are explicitly flagged as a regime where distillation hurts

## Areas for Improvement
- the MLP ablation (Table 4) is only 5 datasets - hard labels outperforming soft labels is a surprising result that goes largely unexplained
- single seed per configuration is a significant weakness
- the 51% win rate framing can be underselling the finding; I personally find the asymmetry (+0.021 on wins vs -0.010 on losses) a more convincing statistic but is buried in the text

## Detailed Comments
- the teacher-rank-to-student-rank finding is clean and practically useful; worth making more prominent

## Justification of Score
- well written paper - easy to read and follow
- OOF insight is the real contribution - simple but important and well-motivated
- large-scale evaluation (153 datasets, 4 teachers, 4 student families) is commendable
- practical guidance is genuinely useful for practitioners
- single-seed evaluation is a notable weakness
- unexplained MLP ablation result (hard labels > soft labels) is concerning

---

### Official Review · Reviewer_tA7j · 2026-05-21
**A mechanism for distilling TFMs**

**Rating:** 5
**Confidence:** 4

**Review:**

Summary

This paper introduces a mechanism for distilling tabular foundation models (TFMs) into GBDT and MLP students, enabling significantly faster test-time predictions for latency-critical applications compared to the original TFMs. The authors identify that, due to the in-context nature of most TFMs, out-of-fold (OOF) labeling over multiple folds is required to generate effective soft teacher labels.

Strengths

1.	Novel insights and methodology: The approach of distilling TFMs into GBDTs using out-of-fold soft labeling successfully avoids the catastrophic collapse of outputs into one-hot vectors.
2.	Robust benchmarking: The authors utilize a comprehensive and robust suite of datasets for evaluation.
3.	Clarity on the viability of findings: The authors transparently state the failure modes and applicable scenarios for their approach.

Areas for Improvement

1.	Performance gains: The performance improvement over the GBDT baselines is marginal, especially considering average rank, which is a more robust metric than average AUC (as average AUC can be easily influenced by outliers).
2.	Baseline selection: The study omits other fast ensemble approaches as baselines. Given that this is a fusion of two tabular prediction paradigms, the inclusion of fast ensemble frameworks like feature stacking (GBDT+MLP) or AutoGluon would strengthen the evaluation.
3.	Experimental scope: The experiments rely on a single seed, and the ablation study regarding setup choices (Section 4.4) is limited to only five datasets.

Detailed Comments

1.	Including a performance comparison against a non-OOF baseline across the full benchmark would better demonstrate the paper's central claims.
2.	The writing style is occasionally abrupt, with limited flow between sentences. The manuscript would benefit from smoother transitions and more cohesive language throughout.

Justification of Score

This paper introduces a novel and pragmatic mechanism for solving the distillation issues inherent in in-context tabular foundation models. However, the marginal performance gains over the student baselines and the restricted use cases (i.e., it is only beneficial for extremely latency-critical applications where the TFM already outperforms baseline students) represent significant limitations.

---

### Official Review · Reviewer_pLzA · 2026-05-21
**Review of the paper: "Pocket Foundation Models: Distilling TFMs into CPU-Ready GBDTs"**

**Rating:** 5
**Confidence:** 3

**Review:**

**Strengths:**

- The paper present a clear problem of TFM: the runtime latency of running these large models on CPU, and address it with a working pipeline of distilling a TFM teacher into a CPU-ready GBDT student
- The empirical finding shows that  OOF labeling is mandatory for ICL teachers, distillation beats CatBoost baseline on 51% of datasets; teacher rank ≈ student rank; gains concentrate on ≤21-feature data; multi-teacher averaging helps MLPs but not trees.
- Extensive evaluations are presented: 153 datasets drawn from four benchmark suites; 4 teachers, 4 student families, and 5 multi-teacher combinations, totaling 48 configurations. The full dataset inventory with sizes and feature counts is provided.
- The paper presets failure modes (distillation ”fails cleanly when the teacher itself underperforms”; high-dimensional tasks in Section 4.2) and clear limitations (”single experimental seed”; “ablation covers one student family on five datasets”)

**Weakness, Questions, and Suggestions:**

- **Single seed per datasets:** Statistically speaking, the results from single seed experiment is not statistically significant and is questionable. I would run the experiment at least five seeds to show ‘some’ statistical significance, and run it with ten seeds or more would be full-paper production level.
- **Lack of background explanation in knowledge distillation and ICL problem:** The authors jump right into the methods before elaborating on the terms such as "out-of-fold label" and "ICL leakage problem." I personally think a clearer and more detailed background information on these problems would help the readers who are not in this domain to understand the problems.
- **Notations:** What is $\tilde{p}$ in the weight in Section 3?
- **Contribution 2 in the Introduction section:** The authors mentioned “Distilling TabICLv2 into XGBoost beats a tuned CatBoost baseline  on 51%……” I am wondering why pick TabICLv2 → XGBoost rather than TabICLv2 → CB with 53% in Table 1’s result?
- **Table 1 and Table 6 results:**
    - In Table 1, the author highlights “TabPFNv2.6→MLP” (.846) as the best method (bolded). However, in Table 6’s extended result of Table 1, we see the results of “LimiX→MLP” (.845) and “TabDPT→MLP” (.846) are similarly good. I wonder if more experiment (multiple seeds) would help distinguish this marginal difference between models.
    - A weaker teacher (TabPFNv2.6) outranks a stronger teacher (TabICLv2) at the MLP student level (Single→MLP), falsifying "no weaker teacher outranks a stronger one in any student family" claim in Section 4.1. How would the authors explain this contradictory result?
- **Table 4 results:** This ablation result confuses me. The Method presented Hinton mixed loss with \alpha=0.7. However, in Table 4, it shows that \alpha=0 is the best. Where is the empirical support for \alpha=0.7 then? Additionally, \alpha=0 means that the soft labels demonstrably don’t help and only the OOF trick does; isn’t this contradicting the premise of "soft labels carry dark knowledge that helps?"
- **Clarification on the confidence-weight function (Eq.1).**: The per-sample weight $w_i = \exp\big(-(H(\tilde{p}_i)-0.7)^2/0.08\big)$ is stated to down-weight both overconfident and near-random teacher predictions, and as a function of the entropy $H$ it attains its maximum at $H(\tilde{p}_i)=0.7$. However, if $H$ is computed with the natural logarithm, a two-class prediction has entropy at most $-(0.5\ln 0.5 + 0.5\ln 0.5)=\ln 2\approx 0.693$, which is below the peak location $0.7$, so on binary tasks $w_i$ increases monotonically as the prediction approaches the maximally uncertain $0.5/0.5$ split. To my understanding, this would up-weight, rather than down-weight, near-random samples, which is the opposite of the stated intent and matters here because the median number of classes across the 153 datasets is two (Table~6). Could the authors specify the logarithm base used for $H$ and confirm whether the target value $0.7$ is appropriate for binary tasks?
- **Inconsistency in dataset size:**
    - Table 2 lists kc2 with d=21 and cmc with d=9, but Table 6 lists 22 and 10 features for the same datasets (Table 2; Table 6). Which one is correct?
    - Table 2 lists internet-ads with n=2,459, while Table 6 lists Internet-Advertisements with 3,279 samples. Is this how the paper is doing train/test split of the samples? Please elaborate on this inconsistency.
    - The abstract mentions “a 38×–860× speedup” while in Section 6 states “850×slower than MLP students.”


Overall, I think this paper is addressing an critical and interesting problem of TFM ICL distillation. However, single seed evaluation and the inconsistency in the numerical results in the writing can be improved. I would be happy to see the outcomes from multiple seed evaluations and a consistent presentation of the results that can address my questions.